# Improving the Monitoring and Management of Clozapine-Induced Gastrointestinal Hypomotility (CIGH) in Community Mental Health Services: A Quality Improvement Approach

**DOI:** 10.3390/pharmacy12050141

**Published:** 2024-09-13

**Authors:** Balazs Adam, Osama Ayad

**Affiliations:** Surrey and Borders Partnership NHS Foundation Trust, Leatherhead KT22 7AD, UK; osama.ayad@sabp.nhs.uk

**Keywords:** clozapine, constipation, CIGH, monitoring, laxatives, community mental health, pharmacy

## Abstract

Clozapine is the only approved antipsychotic for refractory schizophrenia to date. It can cause a range of serious and fatal adverse effects, including Clozapine-Induced Gastrointestinal Hypomotility (CIGH). While guidance is readily available to help manage CIGH effectively in hospital inpatients, practical recommendations applicable to the community (outpatient) setting are lacking. This project set out to improve the prevention, detection and management of CIGH in psychiatric outpatients. An initial baseline audit followed by quality improvement work was undertaken in a busy support worker-run community clozapine clinic focusing on, education and training, risk assessments and clinical documentation. The project was registered and managed using the Life QI web-based platform, where a set of primary and secondary drivers were defined and change ideas were executed. Qualitative and quantitative data were collected over a three-month period, demonstrating a significant improvement in clinical documentation (up from 36% to 99%). 23% of enhanced risk assessments resulted in treatment recommendations, modifiable risk factors were proactively discussed in 53% of clinic appointments and 65% of patients were provided with additional written information on CIGH. It was evident from staff and our patient feedback that further efforts would be required to continue to raise awareness about harms of unmanaged constipation among this client group. Future approaches may include enhanced collaborative efforts with primary care, and improving the skill mix in existing clozapine clinics, which could include the utilisation of mental health pharmacists.

## 1. Introduction

Clozapine is the only approved antipsychotic and evidence-based pharmacological option for treatment-resistant schizophrenia to date [1]. In comparison to other antipsychotics, clozapine has been associated with better concordance, lower suicidal ideation, reduced psychiatric hospital admissions and symptom improvement. [1]. Clozapine remains ’the gold-standard treatment for treatment-resistant schizophrenia‘ [2]. Clozapine has been found to be superior both in terms of quality of life and life expectancy [3] and is listed in the WHO Model List of Essential Medicines [4].

Despite its many benefits, clozapine is not considered to be prescribed widely enough [5]. This is attributed to the many associated serious (and sometimes fatal) treatment risks, including gastrointestinal disorders, glucose intolerance, agranulocytosis and cardiac disorders [6].

Although evidence from pharmacovigilance databases suggest that gastrointestinal complications are the leading cause of clozapine-related deaths [7], they had continued to receive scant attention until the last decade [3].

Clozapine-Induced Gastrointestinal Hypomotility (CIGH) is one of the most common and serious adverse effects of clozapine, with a reported case-mortality rate of 18% [3]. It has been defined as ’an acquired state of delayed transit through the gastrointestinal tract (i.e., transit time > 2 standard deviation above the mean)’ [3].

Constipation—one of the most common features of the CIGH spectrum—is thought to affect 30–60% of patients treated with clozapine [8]. This has historically been viewed by clinicians as a mild adverse effect of antipsychotic therapy. There is now, however, overwhelming evidence that it can lead to paralytic ileus, intestinal pseudo-obstruction, colonic dilatation and perforation, toxic megacolon, sepsis, gastrointestinal ischaemia and necrosis, gastroparesis, feculent vomit and aspiration pneumonia, dehydration, electrolyte imbalance and metabolic acidosis—each with the potential for fatal consequences [9,10].

Colonic transit studies can provide clear objective evidence of CIGH, with an estimated prevalence of 50–80% and mean transit times that are four times longer than normal [11].

In vitro studies have demonstrated that clozapine inhibits neurogenic and—at higher concentrations—myogenic contraction of the colon; this is postulated to be mediated via both antimuscarinic and anti-serotonergic effects [3]. CIGH can affect any part of the gastrointestinal tract, from the oesophagus to the rectum [12]. Although hypomobility may be restricted to distinct sections of the alimentary canal, a cross-sectional study found that 59% of clozapine-treated patients experienced multiregional dysmotility and 89% of the study participants were found to have hypomobility affecting at least one region [13].

Risk factors for developing CIGH have been cited as age, concomitant prescription of constipating agents and a previous history of CIGH; the first four months of treatment are thought to be a particularly hazardous period [7], whereas the risk of ileus appears to be more acute in the late maintenance phases [14]. There are contradictory views on CIGH being a dose-related effect of clozapine [7,9]. With respect to CIGH mortality, male gender, older age, and higher daily doses have been reported as risk factors [7]. Further, a diagnosis of schizophrenia is a known independent risk factor for constipation since many patients will also have sedentary lifestyles, obesity, poor diet as well as low fibre and fluid intake [15].

CIGH has a 10-times higher mortality rate than clozapine-induced agranulocytosis [11,16]. Despite this, CIGH is neither well understood nor well managed [13]. While agranulocytosis-related mortality has been effectively mitigated through haematological monitoring instigated in many countries [12], a similar systemic screening approach has yet to be adopted for the monitoring of CIGH [13,16].

Current approaches to detection often rely on patients’ reporting of constipation—a method shown to have low diagnostic sensitivity (18%) [10]. A key reason for this was cited to be the lack of recognition by patients. This reduced awareness may be caused by negative and cognitive symptoms of schizophrenia, as well as reduced pain sensitivity and ability and/or initiative to communicate discomfort. In addition, clinicians repeatedly fail to spot, assess and effectively treat physical health problems associated with schizophrenia [8,11,17].

The diagnosis of functional constipation is predicated on clinical appraisal due to the lack of universally available objective tests. One such clinical instrument in use to aid diagnosis is the Rome criteria, made up of six key domains [18]. As discussed, however, the CIGH syndrome encompasses many other presentations in addition to constipation. Baptista and colleagues found that 59% of people objectively diagnosed with CIGH via colonic motility testing using radiopaque markers (ROM) were not constipation-positive when assessed using the Rome criteria [19]. This was further corroborated by the diagnostic accuracy study of Every-Palmer et al., who found that while the Rome criteria fared better than patient self-reporting, it only had a modest sensitivity of 48% and a specificity of 63% [10]. Others have advocated for the routine use of stool monitoring charts and adverse-effect rating scales which include constipation [9].

There does not exist one universally accepted and validated treatment protocol for constipation secondary to clozapine. Published data on laxative choice for antipsychotic-related constipation are sparse and early studies were regarded to be of inferior quality [17]. The Porirua Protocol was developed in response so as to help clinicians to manage the symptoms safely and effectively [20].

While guidance on the monitoring and management of CIGH is readily available to organisations caring for clozapine-treated patients, all observational studies discussed above had been conducted in the inpatient setting with access to round-the-clock medical care and other suitably trained staff to perform physical assessments and undertake time-sensitive investigations in response to suspected alarm features. To our knowledge there are no widely adopted guidelines for identifying and effectively managing CIGH in the community (outpatient) setting where such comprehensive care is not routinely available. We therefore set out to scrutinise the activities of one clozapine patient clinic run by one of the busiest community mental health teams in the south of England in relation to the management of CIGH, with the aim of further enhancing practice using a quality improvement approach widely adopted within the National Health Service in the UK.

## 2. Materials and Methods

An initial baseline audit was conducted on the electronic notes pertaining to the team clozapine patient caseload to assess the quality of documentation and to ascertain the extent of existing CIGH monitoring and management in the team as recorded on patients’ clinical records. The results of this work subsequently informed the need for a quality improvement approach to address several concerns highlighted. The project was registered and managed using the Life QI web-based platform.

The primary drivers of the project were staff education and training; psychoeducation of service-users and carers; improved communication and cooperation with, and raising awareness amongst, primary care colleagues; and enhancing the clinical assessment of CIGH.

The secondary drivers included raising awareness and improving recognition of the potential harms; appropriate and timely risk management; and improved documentation and auditability.

The following change ideas were developed for the project:To routinely provide verbal lifestyle advice and information about CIGH to all clozapine patients (and/or their carers);To continually provide easy-to-read patient information leaflets on CIGH to all clozapine patients (and/or their carers);To design and deliver education and training for healthcare workers running the clozapine clinic;To create a patient questionnaire to elicit patient understanding of risks;To design and incorporate standard letters to be sent to GPs with advice to prescribe specific laxatives where appropriate;To develop an enhanced monitoring and assessment form;To design and adopt the use of standardised templates for electronic documentation;To create a written protocol for the new enhanced service.

Data was collected and visualised using run charts to track the project outcome measures centred around quantifying the number of times written information was provided and discussions about risks and lifestyle factors took place with clients. Further, we gathered data on the stratified CIGH risk scores using the enhanced risk assessments. Lastly, we collected feedback from staff and service-users in relation to the problem of CIGH and the enhanced monitoring service. Data collection began in March 2024, and three calendar months’ worth of data had been gathered and analysed by the time of writing.

## 3. Results

### 3.1. Context

The clozapine service, provided by the community mental health team, had been set up as a ‘one-stop shop’ offering all patients prescribed clozapine the convenience to attend for their mandatory blood tests (typically on a four-weekly basis, as specified in the product marketing authorisation) and have their physical health, treatment and side-effects reviewed. Clients would then obtain their clozapine supply from the onsite pharmacy, subject to satisfactory blood test results. The service employed a lead mental health nurse and several support workers trained to provide phlebotomy, processing of blood samples using point-of-care testing, physical health checks and assessment of clozapine-related side-effects using the modified Glasgow Antipsychotic Side-effects Scale for Clozapine (GASS-C), a validated instrument [21]. Patients were reviewed by psychiatrists at set intervals depending on presentation and clinical need, and the service employed a mental health pharmacist who provided advice and guidance, bespoke patient consultations as well as staff education and training.

In relation to CIGH specifically, patients were being asked each time about constipation and their Modified Early Warning Scores (MEWS) were calculated. Information was being captured on preprinted questionnaires which were batch-uploaded to the clinical system as attachments.

The baseline audit conducted on the entire caseload capturing the preceding two-month period highlighted that constipation (and related terms) were only documented for 36% of individual cases. This was attributed to the fact that the questionnaires uploaded by the clozapine team were not themselves searchable documents, and there were inconsistencies among the accompanying searchable electronic entries made by staff members, thus highlighting the lack of universal focus on the problem of CIGH and the absence of standardisation in documentation.

### 3.2. Key Changes

Two education sessions were delivered by the team’s mental health pharmacist where staff learnt about the importance of early recognition and management of CIGH, which also included raising awareness among clients and carers.

To improve documentation and auditability, we created a standardised electronic template to be used routinely by staff running the clinic; this template specifically incorporated CIGH amongst all other monitored parameters and was searchable in the records.

The paper questionnaire used in clinic to note down side-effects and concerns reported by patients (before the information was input into the system) was also updated and the single question pertaining to constipation as part of the GASS-C was replaced by a separate enhanced monitoring and risk assessment form which was developed jointly with the psychiatric consultants in the team; the clozapine team were then trained to administer this appropriately. The form allowed staff to stratify risk with ease to guide them toward taking the appropriate next steps in a clear and concise manner. Three distinct categories of risk were created, ranging from level 1 (minimal risk, to be managed through healthy lifestyle choices), through level 2 (increased risk, to be managed with the addition of laxatives), to level 3 (red flags requiring urgent review). Level 2 was further subdivided based on whether the patient was currently without appropriate laxatives (2A) or whether symptoms were present despite regular laxative treatment (2B); recommendations on treatment were in line with those set by the Porirua Protocol. The monitoring and risk assessment form is available in Appendix A.

A service protocol was implemented to assist staff and clinicians to stratify and manage risk effectively on the day; this included a brief laxative treatment review and the use of new standardised communication to GPs to request further monitoring and treatment in a timely manner. Refer to Figure 1 for a visual overview of the new process.

### 3.3. Quantitative Data

At the start of the project the clozapine caseload was n = 92; this figure increased to n = 110 at the time of writing, which corresponded to three months of data.

Data collection showed the following:Standardised electronic note entries: were made for 285 out of 287 attended clinic appointments over the three-month period (>99% compliance). This demonstrated that staff had adopted the new electronic templates effectively.Lifestyle advice/verbal counselling: was provided during 152 out of 287 attended clinic appointments (53%). Staff had commented that several patients showed signs of cognitive dysfunction and psychomotor retardation, which affected the number of occasions staff felt it useful and/or appropriate to discuss modifiable risk factors in detail.Information leaflets: were handed out in 80 out of 287 attended clinic appointments (28%); these information leaflets were accepted by 72 out of 110 unique patients (65%). Staff had commented that a few patients showed little interest in taking leaflets with them.A total of 269 individual CIGH risk scores were recorded during the set period, corresponding to 90 unique patients; patients excluded consisted of those who did not consent to their scores being calculated (declined to answer questions) on at least one occasion, as well as one patient who had undergone colostomy following diagnosis of bowel cancer. 23% of enhanced assessments yielded a risk score above 1, resulting in treatment recommendations. See Figure 2 for the breakdown of the risk scores among the cohort. Individual variability in the cohort over the data collection period is visually represented in Figure 3.

### 3.4. Qualitative Data

We gathered informal verbal feedback from staff at baseline and at three months. At both time points staff overwhelmingly expressed enthusiasm about the enhancements introduced. Some reflected about patients’ reception and comments, namely that some patients were expressing reluctance to discuss bowel habits. This may explain the relatively large proportion of patients consistently reporting symptoms corresponding to level-1 CIGH risk score. Another distinct theme of staff feedback pertained to the added time (and thus workload) the new process presented. Despite this, the team were in favour of continuing the enhanced service, expressing their commitment to being proactive in their approach when discussing CIGH risks and management options, including healthy lifestyle choices, with patients, and focusing on tactfully eliciting the extent of the problem.

Comments were also collected from patients. At the three-month mark the team handed out questionnaires to learn about patients’ understanding of CIGH risks and to gauge the interest among patients for another potential service improvement initiative in future which would entail the provision of appropriate laxative treatments in the clozapine clinic. By the time of writing, 14 questionnaires had been returned by patients; 13 responded that they were aware of constipation being a side-effect of clozapine, and 13 confirmed having received written information about associated risks from staff. Six patients provided brief additional information pertaining to perceived risks of constipation: four responded with “blockage”, of which one also stated “cancer”; one answered, “just know that it can cause constipation which is obviously bad for you”, and one wrote, “unsure about the risk”. With respect to the service proposal for the provision of complimentary laxatives, 12 were ‘positive’ or ‘very positive’, one was ‘neutral’, and one was ‘negative’ with the additional comment that they were “concerned about having extra medication that would not be able to come off”. One patient also commented, “good service”.

## 4. Discussion

As presented earlier, despite the pronounced risks of CIGH, the problem remains largely overlooked. Published observational studies on CIGH are predicated on inpatient data [10,20]. There is a paucity of established clinical guidance on how best to manage CIGH in the community setting, where psychiatrists may no longer possess relevant up-to-date competencies to undertake a comprehensive (physical) assessment of associated symptoms and there may not be other suitably trained staff to provide these. GP feedback sought informally suggested that while GPs might be competent at managing constipation symptoms in the general population, understanding about CIGH - specifically its management - was poor amongst primary care colleagues. Since our clozapine patients were mostly seen in clinic by healthcare workers rather than by registered healthcare professionals, we were also cognisant of the knowledge and skills gap in recognising and thoroughly assessing constipation and other features and risks associated with CIGH. While some of the signs of an acute deterioration—such as moderate or severe abdominal pain lasting over an hour, abdominal distension, vomiting or bloody diarrhoea—would be relatively simple to ascertain, specific signs of sepsis and other indicators of severe harm—such as absent or high-pitched bowel sounds, metabolic acidosis, haemodynamic instability and leucocytosis—would require careful assessment by specially trained staff [9,20]. Additionally, we recognised that suggestions made by experts, such as consideration for a medical abdominal examination in the absence of bowel movement over the preceding three days [9], or a digital rectal examination following an unsuccessful 48 h trial with laxatives [20] would seldom be feasible in the outpatient setting. Our collaborative work with primary care providers highlighted that these proposed timelines could also rarely be met in primary care, owing to the intense strain and current work pressures on the National Health Service; this made the patient education component on the importance of patients recognising and reporting significant changes without delay particularly salient.

It was evident from our patient feedback that continued efforts were necessary to raise awareness about the harms of unmanaged constipation among this client group. This was corroborated by a large volume of previous studies emphasising the need for increasing awareness and education amongst both patients and healthcare staff, including the importance of also proactively addressing modifiable risk factors [7,8,16,22].

It has already been demonstrated that mental health pharmacists can play a vital role not only in overseeing the appropriate supply of clozapine, but also in education and training of staff, as well as in the comprehensive monitoring of this high-risk drug [23,24]. In the case of this particular project, in-house training of staff was delivered by the team’s mental health pharmacist, focusing on the CIGH spectrum as well as its management including lifestyle changes and medication. Staff attitudes to continuous learning and delivering quality improvement also played a key role in the implementation of the project. The clozapine team continually expressed their commitment to providing proactive patient education, and we also had the opportunity to incorporate into the project the provision of easy-to-read information leaflets on the topic, already available to the organisation via subscription (Choice and Medication, Mistura Enterprise Limited).

Insights from staff were invaluable in interpreting the CIGH risk scores obtained through consultations with patients. As Every-Palmer and colleagues had suggested earlier, patients taking clozapine may lack the ability to recognise the symptoms due to several factors put forward [10]. We found that many of our patients were reticent to discuss their bowel habits and potential gastrointestinal problems, despite established relationships with clinic staff. It was conjectured that several patients might not have fully appreciated the reasons for the apparent sudden change in focus by the clinic team on this adverse effect and associated risks after years (in some cases, decades) of them taking clozapine. Again, this reinforces the need for continuous education in order to overcome the barrier of hesitancyand stigmaaround discussing bowel habits.

While the newly designed monitoring and assessment form was not intended to be a validated instrument for identifying CIGH and all related red-flag symptoms (since this was outside the scope of this project), we endeavoured to produce a useful tool that allowed for the active exploration and candid discussion of associated features and risks. Although the validated GASS-C tool was already in use by the team, we undertook to enhance the assessment process based on published evidence suggesting that screening for constipation in clozapine patients using a single question significantly lacked diagnostic sensitivity [10]. We had considered the works by others who had indicated that stool frequency alone was not sufficient for identifying constipation [25], and that stool consistency better correlated with colonic transit time [26]. Many studies had deemed the Bristol Stool Chart—one of the most widely used tools in both clinical care and research—to be particularly useful in determining stool form [27,28]. We followed the advice of some publications asserting that measuring both frequency and consistency might better predict overall mortality risk [29], which is a fundamental consideration for people with schizophrenia, who are known to die younger than the general population or those with any other psychiatric disorder [30]. Though not an externally validated tool, the Porirua Protocol is widely recognised and accepted among experts [2,9,20], and hence we incorporated it into our new processes and directly referred to it in our written recommendations to primary care colleagues. Further, we included the consideration for colon cancer as recommended by previous published work [31]. In designing the monitoring and assessment form we were careful to only include concepts and terminologies appropriate for the competency level of healthcare support staff running the clinic. Therefore, while it had been recommended that the Rome criteria should form the basis of any tool used to assess constipation [32], the inherent complexity of the individual criteria meant that this was not appropriate for adoption in this setting. To mitigate for the lack of specialist skills and knowledge, we worked closely with the clozapine team to agree upon the vital nature of rapidly escalating concerns to the medical staff on duty who could apply further clinical judgement on an individual basis.

With respect to the new monitoring and assessment form, the clinical team carefully considered the inclusion and weighting of the many factors at play to create a scoring system that guided patients and healthcare professionals alike to managing the condition quickly and appropriately in this support worker-run clinic. We were also acutely aware of current work pressures and had to carefully balance the clinical need for laxatives with the practicalities of potentially referring a large volume of patients to their GP. We propose that in other settings, such as those run by physical health nurses or pharmacists, the assessments could entail additional aspects recognised to influence the extent of CIGH. For instance, age as a risk factor [33] (particularly in services looking after older adults) and the results of relevant physical examinations [9,20] could also be considered in the risk-stratification and overall decision-making process. Particularly for mental health pharmacist-led clinics, the inclusion of anticholinergic burden of concomitant pharmacotherapy [8,12,33,34] may be of value. Adherence to laxative treatment [33] and—where osmotic laxatives are being used long term—the monitoring of electrolytes including magnesium levels [9] would also be useful. A universal move towards the prophylactic use of stimulant and softening agents for all clozapine-treated patients, irrespective of symptoms, would be of particular benefit in terms of harm minimisation [9,10]. Although the systems-wide adoption of prophylactic laxatives will require a deep cultural shift within organisations, many experts continue to advocate for this based on similar approaches that have been widely adopted in other clinical scenarios, such as in the case of preventing constipation secondary to opiates [32,35]. The authors assert that adopting this approach could be smoother if rolled out initiallyin the inpatient setting where a larger number of clozapine initiations tend to take place and a team of different professionals are on hand to provide round-the-clock care and advice to patients. Alternatively, as indicated by the feedback we received from our patients, there may also be an interest in the implementation of such a service that provides routine supply of laxatives, be it via prescriptions, or through other legal mechanisms, such as patient-group directions, which is one approach already adopted by other mental health trusts with success.

### 4.1. Limitations

Working on the project we were met with several challenges. While we were able to demonstrate that simple changes within an established team could deliver sustainable improvements to patient care, universal work pressures and a lack of physical health nurses or other suitably trained professionals in the service meant that there was a knowledge and skills gap in providing detailed assessments, thus rendering the comprehensive management of CIGH in-house impracticable and necessitating collaboration with primary care. Standard letters were sent to the GP for patients categorised as risk level 2 (A & B); however, no confirmations of receipt or acceptance were received. This highlighted the necessity for implementing better systems to facilitate effective two-way communication with GPs. Further, some patients with red-flag symptoms required direct referral to acute care, which indicated a need for close cross-sector collaboration with gastrointestinal specialists to create appropriate pathways of care. We concluded that ultimately there was a need for regional and/or national strategic initiatives at the health systems level to make such partnership working effective and viable.

Many of the patients studied showed evidence of cognitive dysfunction due to both the chronicity of their psychotic illness and the long-term use of clozapine. This limited the usefulness of discussing modifiable risk factors in detail in clinic. In addition, several patients declined to accept information leaflets and some of them also refused to engage with the new risk assessment process. We anticipate that the engagement rate will improve as staff continue to discuss the risks of CIGH and the importance of monitoring to patients.

### 4.2. Conclusions

In this work we underlined the vital importance of appropriate skill mix, and interdisciplinary and cross-sector cooperation so as to improve the prevention, detection and appropriate management of CIGH for psychiatric outpatients. We proffer several sustainable solutions to this, including an enhanced monitoring, assessment and risk stratification tool; systems for standardised documentation and correspondence with GPs; as well as staff training, upskilling and proactive patient education. Most of these can be readily implemented and embedded into community mental health services, including Early Intervention In Psychosis teams, without additional resources; however, long-term effective management will likely require a dedicated workstream. We posit that pharmacist-led clozapine clinics could fill the gap in enhanced service provision by providing detailed assessments and monitoring, as well as offering treatments for the many side-effects of clozapine, including but not limited to constipation. Future works evaluating such services would likely provide vital insights and evidence to support commissioning decisions.

## Figures and Tables

**Figure 1 pharmacy-12-00141-f001:**
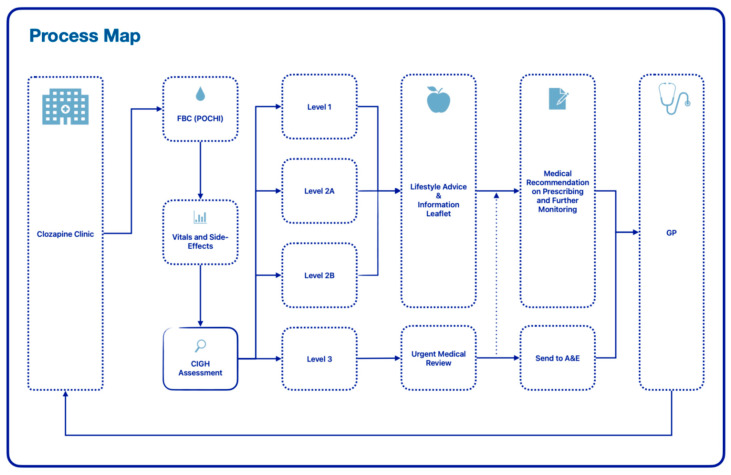
Process map for enhanced service.

**Figure 2 pharmacy-12-00141-f002:**
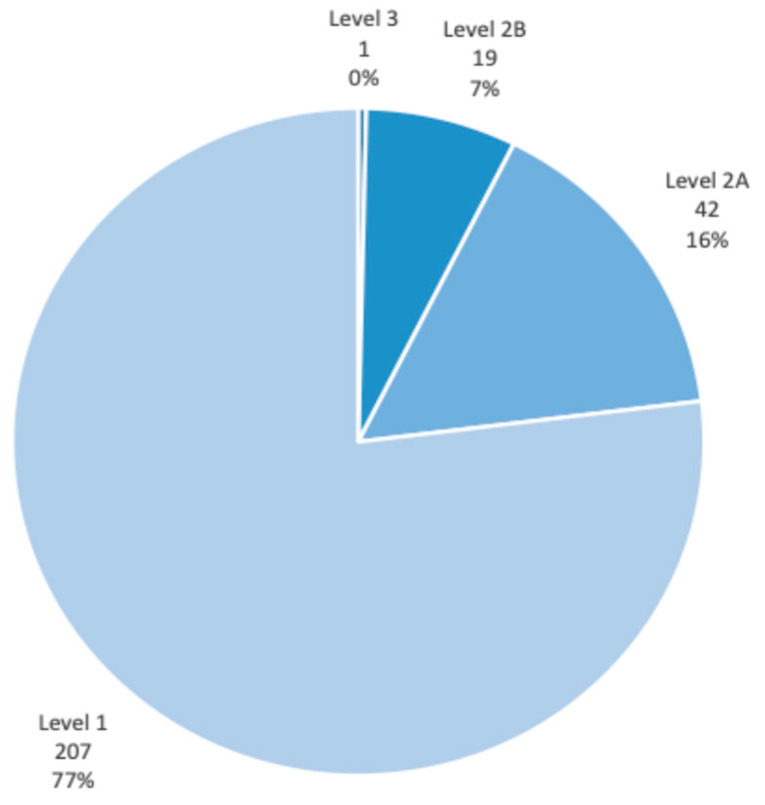
CIGH risk score distribution.

**Figure 3 pharmacy-12-00141-f003:**
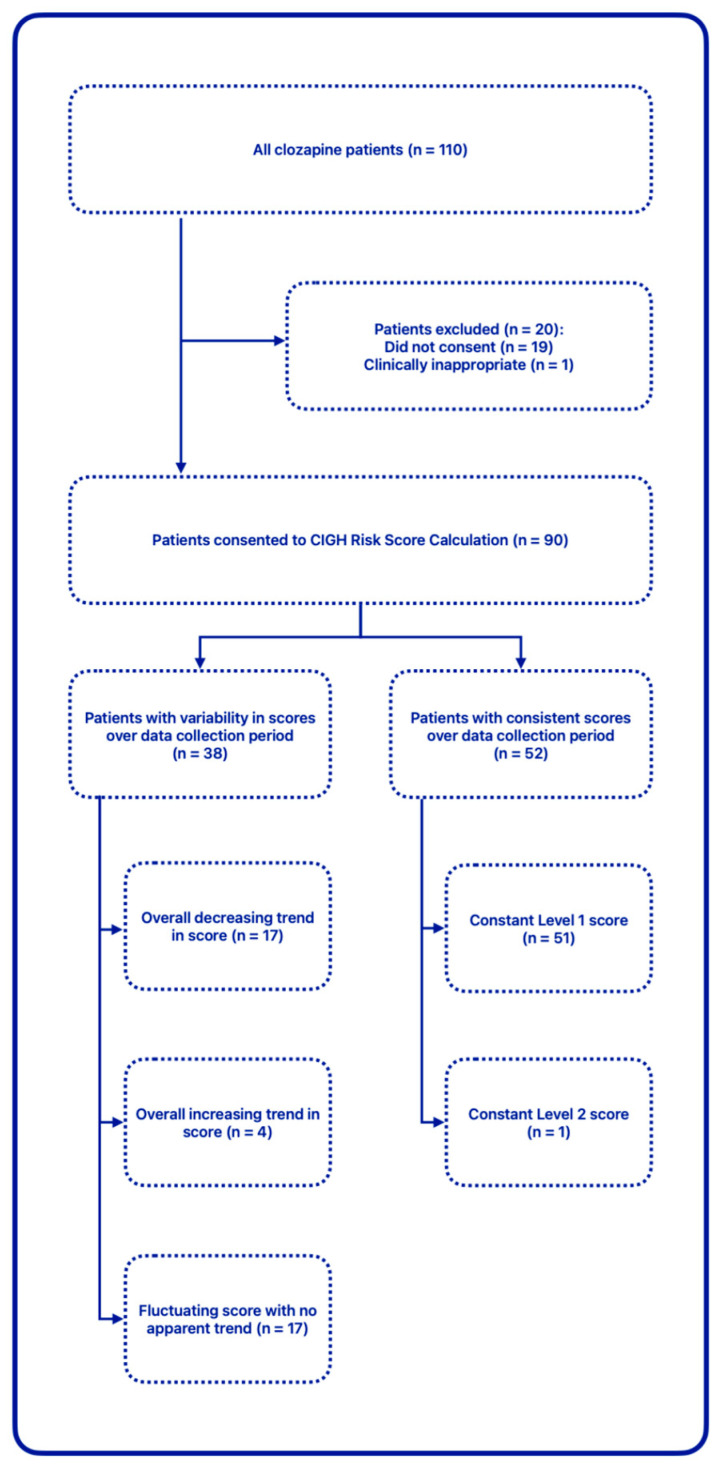
CIGH risk score variability.

## Data Availability

The raw data supporting the conclusions of this article will be made available by the authors on request.

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
