# Peer review of "Improving the Monitoring and Management of Clozapine-Induced Gastrointestinal Hypomotility (CIGH) in Community Mental Health Services: A Quality Improvement Approach"

_pharmacy, 2024, doi:10.3390/pharmacy12050141_

Round 1

Reviewer 1 Report

Comments and Suggestions for Authors

I read with interest the paper titled "Improving the monitoring and management of Clozapine-Induced Gastrointestinal Hypomotility (CIGH) in community mental health services: a quality improvement approach"

1. Authors refer the study as a Quality Improvement Study. My suggestion is that authors use the SQUIRE guidelines for reporting the manuscript, which will improve (a lot) the paper. Please check https://www.equator-network.org/reporting-guidelines/squire/

2. Please check the guidelines of the journal when reporting bibliography. 

3. Bibliography is quite lenghty. Please reduce it and keep it simple, focusing on the objective of your paper. 

4. The objective should be clearly stated in the abstract. Please clarify. 

5. I suggest to present some of the results in table/figures, instead of text. A lot of text is usually difficult to read and interpret (for eg: quantititive data).

6. Please adress the limitations of your study in a different subchapter of the discussion. 

7. I believe your paper is well discussed. However, please create a clear conclusions with the following topics: a.  Usefulness of the work; b.  Sustainability; c.  Potential for spread to other contexts; d.  Implications for practice and for further study in the field; e.  Suggested next steps. This is in line with SQUIRE guidelines, which I believe will improve a lot the readiness of your paper. 

Author Response

I believe your paper is well discussed. However, please create clear conclusions with the following topics: a. Usefulness of the work; b. Sustainability; c. Potential for spread to other contexts; d. Implications for practice and for further study in the field; e. Suggested next steps. This is in line with SQUIRE guidelines, which I believe will improve a lot the readiness of your paper.

Thank you for pointing this out. We agree with this comment and have made the following changes: A conclusions subchapter has been created following the recommendations from SQUIRE. (page 11, line 511)

Please address the limitations of your study in a different subchapter of the discussion.

Thank you for pointing this out. We agree with this comment and have made the following changes: Limitations have now been added as a subchapter. (page 11, line 490)

I suggest to present some of the results in table/figures, instead of text. A lot of text is usually difficult to read and interpret (e.g.: quantitative data).

Thank you for pointing this out. We agree with this comment and have made the following changes: Both the CIGH risk score distribution and variability are now represented as figures only. (page 7, line 283; page 8, line 286)

The objective should be clearly stated in the abstract. Please clarify.

Thank you for pointing this out. We agree with this comment and have made the following changes: A sentence has been added to explicitly state the objective. (page 1, line 15)

Bibliography is quite lengthy. Please reduce it and keep it simple, focusing on the objective of your paper.

Thank you for pointing this out. We agree with this comment on general principle, however we have only been able to reduce the bibliography by one as we feel all citations were relevant in not only describing the QI work itself, but also the extent of the problem of CIGH.

Please check the guidelines of the journal when reporting bibliography.

Thank you for this - we have updated the referencing style accordingly.

Authors refer the study as a Quality Improvement Study. My suggestion is that authors use the SQUIRE guidelines for reporting the manuscript, which will improve (a lot) the paper. Please check https://www.equator-network.org/reporting-guidelines/squire/

Thank you for this excellent resource. We have taken SQUIRE into account as we restructured the presentation of the manuscript, particularly the Discussion section. (page 9, line 287)

Reviewer 2 Report

Comments and Suggestions for Authors

This study is focussed in the monitoring and management of gastrointestinal problems produced by Clozapine. 

It is interesting to have ways to avoid these adverse effects that could lead to severe problems for the patient

There manuscript is correct but there are some considerations that I would like to point out:

-Introduction:

·        Even though the authors indicate a reference for the prevalence of constipation, it could be interesting to have also the information included in the VigiAcess of WHO in relation to this adverse effect.

·        In order to understand better the whole process, the authors could explain a little more about how work the community mental health teams

-      Results:

I would recommend Appendix B as a Figure of the text, as it is very informative.

-      Discussion:It could be a little shorter and include a Conclusions section

Author Response

Introduction: Even though the authors indicate a reference for the prevalence of constipation, it could be interesting to have also the information included in the VigiAcess of WHO in relation to this adverse effect.

Thank you for pointing this out. While we agree that pharmacogivilance data are vital, these adverse effects are discussed extensively in this work with incidence data from primary literature. Databases have differing reporting rates, including VigiAccess (WHO), MHRA DAPs (Medicines and Healthcare products Regulatory Agency, UK) and we feel that the inclusion of any of these is of limited benefit. Taking into consideration the other peer reviewer’s recommendations to reduce the overall length and bibliography, we have decided not to add VigiAccess data, as we feel we have emphasised the extent of the problem of CIGH sufficiently throughout the manuscript.

Introduction: In order to understand better the whole process, the authors could **explain a little more about how work the community mental health teams.

Thank you for pointing this out. This is now further expanded upon in 3.1 Context. (page 4, line 166)

Results: I would recommend Appendix B as a Figure of the text, as it is very informative.

Thank you for this. Appendix B is now Figure 1. (page 6, line 278)

Discussion: **It could be a little shorter and include a Conclusions section.

Thank you for this. Although we have not been able to shorten the section, it has been reorganised to improve readability, and a Conclusion subsection has also been added. (page 11, line 511)